# *Drosophila melanogaster*: A Model System to Study Distinct Genetic Programs in Myoblast Fusion

**DOI:** 10.3390/cells11030321

**Published:** 2022-01-19

**Authors:** Pratiti Rout, Mathieu Preußner, Susanne Filiz Önel

**Affiliations:** 1Molecular Embryology, Department of Biology, Philipps University Marburg, Karl-von-Frisch Str. 8, 35043 Marburg, Germany; routp@staff.uni-marburg.de; 2Membrane Plasticity in Tissue Development and Remodeling, DFG Research Training Group, GRK 2213, Philipps-University Marburg, 35037 Marburg, Germany; 3Developmental Biology, Department of Biology, Philipps University Marburg, Karl-von-Frisch Str. 8, 35043 Marburg, Germany; preussner@bio.uni-frankfurt.de

**Keywords:** cell–cell communication, F-actin, cell–cell fusion, membrane fusion, visceral muscles, somatic muscles, testis muscles, flight muscles

## Abstract

Muscle fibers are multinucleated cells that arise during embryogenesis through the fusion of mononucleated myoblasts. Myoblast fusion is a lifelong process that is crucial for the growth and regeneration of muscles. Understanding the molecular mechanism of myoblast fusion may open the way for novel therapies in muscle wasting and weakness. Recent reports in *Drosophila* and mammals have provided new mechanistic insights into myoblast fusion. In *Drosophila*, muscle formation occurs twice: during embryogenesis and metamorphosis. A fundamental feature is the formation of a cell–cell communication structure that brings the apposing membranes into close proximity and recruits possible fusogenic proteins. However, genetic studies suggest that myoblast fusion in *Drosophila* is not a uniform process. The complexity of the players involved in myoblast fusion can be modulated depending on the type of muscle that is formed. In this review, we introduce the different types of multinucleated muscles that form during *Drosophila* development and provide an overview in advances that have been made to understand the mechanism of myoblast fusion. Finally, we will discuss conceptual frameworks in cell–cell fusion in *Drosophila* and mammals.

## 1. Introduction

The movement of vertebrates is controlled by bundles of multinucleated skeletal muscle fibers that represent one-half of the body’s mass. Each myofiber comprises thousands of nuclei and arises by the continual fusion of mononucleated myoblasts during embryogenesis. However, the fusion of myoblasts is also crucial after birth for the growth and repair of muscles. Therefore, understanding the mechanism of myoblast fusion during development has a major impact on the treatment of muscle wasting in adults, such as aging, muscle atrophy, or dystrophy [1]. Many investigations into myoblast fusion have been carried out in the fruit fly *Drosophila melanogaster*. Important advantages of using *Drosophila* as a model system are the short life cycle and the powerful genetics to unravel gene function. Furthermore, 60% of the *Drosophila* genome is conserved to that of humans [2].

The fusion of myoblasts is a special type of membrane fusion, whereby two lipid bilayers merge to become one [3]. Well-characterized types of membrane fusion are the fusion of organelles and the fusion of pathogens with host cells [4]. Studies on organelle and virus fusion have established that membrane fusion is an energy-consuming process. The energy barrier of this process must be high enough to avoid spontaneous membrane fusion, but must be low enough so that specialized proteins, so-called fusogens, are able to conquer this energy barrier [3]. But what determines the heights of the energy barrier? Clearly, the lipid composition of the membrane is important. However, the energy barrier of the membrane is further influenced by mechanical stress, such as curvature and lateral membrane tension. Membrane tension can arise by osmotic pressure, from forces caused by cell adhesion, and from membrane interactions with the underlying actin cortex. Studies in *Drosophila* have established that the dynamic rearrangement of the actin cytoskeleton is essential for myoblast fusion [5]. At the site of cell–cell contact, a transient cell–cell communication structure forms that involves the ring-like expression of cell adhesion molecules of the immunoglobulin family in which an F-actin-rich center forms. This cell–cell communication structure has been termed fusogenic synapse (also known as fusion-restricted myogenic adhesive structure, shortened as FuRMAS, or podosome-like structure, shortened as PLS). On ultrastructural level the formation of finger-like protrusions has been reported and it has been suggested that the invading protrusions and the actomyosin cortex provide membrane bending stress and lateral tension for fusion pore formation and expansion [6].

The ring-like expression of cell adhesion molecules at the site of cell–cell contact triggers the formation of branched F-actin and the recruitment of electron-dense vesicles, which finally leads to the fusion of the myoblasts. This structure has been termed fusogenic synapse.

During embryonic and adult development, myoblasts are specified and undergo changes in their gene expression profile while fusion proceeds. In fact, the fusion of myoblasts must be spatially and functionally restricted. Although many transcription factors are known to contribute to muscle tissue formation, not many epigenetic mechanisms like DNA methylation, histone modifications, or micro-RNAs, have been discovered in myoblast fusion, to date.

In the first part of this review we will introduce the different *Drosophila* syncytial muscle-types. Subsequently, we will give an overview of different studies that have been conducted during *Drosophila* embryonic and adult muscle development to unravel the mechanism of myoblast fusion. We will highlight common and different players directing myoblast fusion in different *Drosophila* muscle-types as well as in skeletal muscles.

## 2. Muscle Formation during *Drosophila* Development

*Drosophila* is a holometabolic insect that shows major morphological differences between larval and adult stages. Consequently, the larval muscles that have been created during embryogenesis are destroyed during metamorphosis at pupal stages and are replaced by adult muscles. In the embryo, the formation of the larval bodywall muscles has been studied extensively. In contrast to mammalian muscles, *Drosophila* bodywall muscles are built by one muscle fiber per muscle. Furthermore, the somatic muscles are organized in a very simple pattern of 30 segmentally repeated muscle fibers. Whereas mammalian muscles contain up to thousand nuclei, the somatic muscles of *Drosophila* incorporate 4 to 24 nuclei. Besides somatic muscles, the visceral muscles of *Drosophila* surrounding the gut are syncytial. The larval midgut muscles are composed of binucleated circular muscles that show an incomplete fusion and of multinucleated longitudinal muscles.

In the adult, formation of the flight muscles has been studied predominantly. The flight muscles of *Drosophila* are more related to mammalian skeletal muscles and contain up to 1000 nuclei. They arise from quiescent myogenic precursor cells that have been determined during embryogenesis and are located on the imaginal disc, the so-called wing imaginal disc. During metamorphosis, a set of larval muscles persists through histolysis and serves as a template for development of a subset of flight muscles, called Dorsal Longitudinal Muscles (DLM). The template muscles fuse with myoblasts derived from the myogenic precursor cells. Interestingly, in a recent study, quiescent satellite cells were identified in the flight musculature, which can be, like in mammals, activated by injury. A further informative system to study adult myoblast fusion is the formation of the testes muscles. Like adult flight muscles, the testes muscles develop from myogenic precursor cells that are localized on the genital imaginal disc. However, unlike the myogenic precursor cells that differentiate into myoblasts through their migration towards the template muscles, these cells fuse on the genital imaginal disc after differentiation. An overview of these different muscles types is depicted in Figure 1.

### 2.1. The Formation of the Larval Bodywall Muscles during Embryogenesis

#### 2.1.1. Founder Cells and Fusion Competent Myoblasts

In *Drosophila* two populations of myoblasts undergo fusion: founder cells and fusion-competent myoblasts. Both myoblast types derive from areas of the mesoderm that express high levels of the bHLH transcription factor Twist. Although Twist is expressed in the entire mesoderm, the difference in the level of Twist expression determines the fate of the somatic and cardiac muscles that arise from areas of high Twist expression. The fate of the visceral muscles and the fat body emerges in areas of low levels of Twist [7]. Within the high Twist domain the transcription factor Lethal-of-Scute defines a myogenic cluster [8]. In this cluster a muscle progenitor cell is singled out by a cross talk between the Notch and Ras/MAPK signaling pathway [9]. These progenitor myoblasts divide asymmetrically and give either rise to two founder cells or to a founder cell and an adult precursor cell (AMP). The latter will remain quiescent and stay undifferentiated during embryonic life, but will be reactivated during second larval instar to generate muscles of the adult fly [10]. The remaining cells of the myogenic cluster that express Notch develop into fusion-competent myoblasts [9]. The founder cells define the unique identity of the muscle fiber, e.g., its size, shape, orientation, and attachment site and express a specific combination of identity transcription factors by which they can be identified [11]. On the contrary, the identity of the fusion-competent myoblasts is not unique. They express the Gli transcription factor Lame duck (Lmd) [12]. As described in the following sections, fusion-competent myoblasts can fuse with different pools of founder cells. After the fusion of a founder cell with a fusion-competent myoblast, the fusion-competent myoblast usually adopts the fate of the founder cell and the nuclei of the founder cell express the unique identity transcription factors of the founder cell [13]. How the reprogramming of the fusion-competent myoblast occurs is not known yet.

#### 2.1.2. Cellular Steps of Embryonic Myoblast Fusion

By combining light and electron microscopy studies, different morphological steps of myoblast fusion could be distinguished. The first step to initialize the fusion process is the recognition and adhesion of both cell types. On light microscopy level, fusion-competent myoblasts can be recognized by their ability to form a teardrop-like shape (Figure 2A, asterisks). Immunohistochemical analyses and life imaging studies have revealed that the adoption of this shape is required for the migration towards the founder cell and the recognition of both cell types [14,15,16]. After myoblast recognition, the adhesion of both myoblast types is intensified (Figure 2B). On subcellular levels, distinct ultrastructural features (Figure 2C) have been described by using chemical fixation and freeze substitution methods [16,17,18,19,20,21,22]. However, the temporal appearance of some of these structures is still unknown (Figure 2(C1–3)). Following recognition and adhesion, electron-dense plaques and vesicles have been observed at the site of cell–cell contact (Figure 2(C1,C2)). About 50 of these vesicles can be found in adhering myoblasts where they line up with one another to form pairs across the apposing membranes [17,23]. The function of these vesicles remains unknown. The electron-dense plaques are reminiscent of cadherin-containing cellular junction, e.g., adherens junctions [22]. During the last decade the formation of finger-like protrusions has been discovered (Figure 2(C3)). Their formation and function are in the focus of current investigations in embryonic myoblast fusion [21,24,25]. Lastly, a fusion pore is formed leading to the mixing of the content of both myoblast types and to the integration of the fusion-competent myoblast into the founder cell (Figure 2(C4)). The fusion of myoblast finally results in the formation of a segmentally repeated muscle pattern (Figure 2D).

#### 2.1.3. Establishment of a Cell–Cell Communication Structure after the Recognition and Adhesion of Myoblasts

Classical and molecular genetics have revealed that the recognition and adhesion of founder cells and fusion-competent myoblasts depends on myoblast-type specific cell adhesion molecules of the immunoglobulin superfamily IgSF [13,23]. Dumbfounded (Duf/Kirre) and its homolog Roughest (Rst) are expressed in founder cells where they serve redundant functions [26]. Hibris (Hbs), Sticks and Stones (Sns), and Roughest (Rst) are found in fusion-competent myoblasts [27,28]. Duf, Sns, and Rst are expressed in a ring-like structure at cell–cell contact points [29]. The interaction of Duf and Sns leads to the formation of a transient ring-like cell–cell communication structure known as FuRMAS/podosome-like structure and recruits downstream proteins, filamentous (F-) actin, and electron-dense vesicles and plaques to the site of cell–cell contact (Figure 3) [14,21]. This structure is similar to other cell–cell communication structures, like the immunological synapse that forms between the interface of a T-cell and an antigen-presenting cell [29,30]. The interaction of T-cell receptors with major histocompatibility complex (MHC) proteins on the site of the antigen-presenting cell (APC) leads to co-receptor engagement, phosphorylation of downstream signals, formation of F-actin, organization of microtubules that transport intracellular vesicles, and translocation of mitochondria [31].

In addition to the members of the IgSF, the cell surface receptor N-cadherin is expressed in founder cells and fusion-competent myoblasts during the process of myoblast fusion [32]. Although the role of N-cadherin during *Drosophila* myoblast fusion is still not solved, a persistence of N-cadherin expression in myoblasts that undergo fusion seems to prevent myoblast fusion. The removal of N-cadherin from the plasma membrane involves the activity of the guanine-nucleotide exchange factor Schizo/Loner [32], which is antagonized by Abi (a member of the WAVE regulatory complex) [33]. Members of the IgSF and cadherin superfamily are also involved in mammalian myoblast fusion [34]. However, their role for the fusion process is not as well defined as in *Drosophila*.

#### 2.1.4. Intracellular Signaling and Branched F-Actin Formations during Myoblast Fusion

After the formation of the ring-like cell adhesion structure, intracellular proteins are recruited towards the side of cell–cell contact resulting in extensive actin remodeling in founder cells and fusion-competent myoblasts (Figure 4). In founder cells, a thin actin sheath forms which is thought to provide the tension for the fusion process. By contrast, a prominent F-actin focus of approximately 2 μm^2^ is formed in fusion-competent myoblasts. The cytodomains of the IgSF proteins contain phosphorylated tyrosines and proline-rich motives that create an interaction surface for SH2 and SH3 domain containing proteins [35,36]; e.g., Dreadlock (Dock) and Crk. These adaptor proteins translate the fusion signal from the plasma membrane into the cell resulting in the activation of the evolutionary conserved Arp2/3 complex. The activation of the Arp2/3 complex involves nucleation-promoting factors (NPFs) of the Wiskott–Aldrich Syndrome protein family.

*WASp*: A dominant-negative *wasp* allele showing severe defects in myoblast fusion was identified in an EMS mutagenesis screen [37]. In hypomorphic *wasp* alleles, the high maternal contribution of the *wasp* mRNA compensates for the loss of *wasp* and myoblast fusion defects are only visible in maternal and zygotic mutants [37]. In mammals, WASp associates with Wip. The release and degradation of WASp from this complex is triggered by the phosphorylation of Wip [38]. In *Drosophila*, Wip is exclusively expressed in fusion-competent myoblasts where it localizes to the site of cell–cell contact [18,20,39]. Another fusion-relevant protein that can be found solely in fusion-competent myoblasts is the PH domain containing protein Blown fuse (Blow) [14,17]. After the recognition and adhesion of myoblasts, Blow is recruited to the site of cell–cell contact where it localizes with F-actin as a dense focus [14,36]. Blow possesses two Wip interaction domains and competes with WASp for Wip binding [36]. Whether in *Drosophila*, the release of the Wip-WASp complex is triggered by the phosphorylation of Wip like in mammals is unknown.

Another factor that is involved in the fast elongation of filaments and thus might be able to contribute to the formation of finger-like protrusions is WHAMY. WHAMY belongs to the Wiskott–Aldrich Syndrome protein family and originated from a *wasp* gene duplication. But unlike WASp it cannot activate the Arp2/3 complex [40]. Homozygous *whamy* mutants are viable, but show severe myoblast fusion defects in combination with a hypomorphic *wasp* allele.

*WAVE*: WAVE forms a complex with Abi, HSP300, Sra1, and Kette (known as Nap1/Hem1 in mammals) [41]. Like *wasp*, it possesses a high contribution of maternal mRNA and defects in myoblast fusion are only visible in maternal and zygotic mutants [42]. The activity of the WAVE complex is regulated by the binding of activated Rac to Sra1 [43]. The binding of Rac leads to the exposure of the Arp2/3 binding domain by a conformational change Sra1–Kette subcomplex [44]. Consequently, *rac1* and *rac2* double mutants and *kette* single mutants show impaired myoblast fusion [45]. The activation of Rac depends on the guanine nucleotide exchange factor *myoblast city* (*mbc*) [46]. Another factor that acts together with Mbc and contributes to the activation of Rac is Elmo [47]. Activated Rac was shown to interact with *Drosophila* Pak3 (DPak3) [48]. DPak3 belongs to the family of p21-activated kinases, which serve as Ser/Thr kinases. Although DPak3 serve redundant functions with DPak1 in myoblast fusion, DPak3 has been reported to play a major role in myoblast fusion. Furthermore, it is enriched at the site of fusion at cell–cell contact sites where it colocalizes with F-actin in fusion-competent myoblasts [48]. The binding of activated Rac1 leads to the autophosphorylation of Pak and to the phosphorylation of downstream target [49].

*Phosphatidylinositol 4,5-bispohosphat*: The Rac activator Mbc is recruited by phosphatidylinositol-4,5-bisphosphat (PIP_2_) during myoblast fusion. In embryos expressing the pleckstrin homology domain of phospholipase C-gamma, Mbc fails to concentrate at the site of fusion. The mislocalization of Mbc also leads to the mislocalization of downstream targets, e.g., WAVE [50].

*Diaphanous*: In addition to Arp2/3-dependent F-actin formation, linear F-actin formation nucleated by Diaphanous (Dia) plays a role in myoblast fusion [51]. Dia is enriched at the F-actin focus and either the loss or the overexpression of activated Dia blocks myoblast fusion.

*Further cell cortex proteins*: An important element of the plasma membrane–cortex composition is Spectrin. This protein is able to assemble into a nonpolarized meshwork that is connected the plasma membrane, the actin cytoskeleton, and associated proteins. The α/β_H_-Spectrin heterotetramer is only required in founder cells where it colocalizes with Duf [52]. Although it localizes independently of Duf to the site of cell–cell contact, the dispertion of Duf was observed in α/β_H_-*spectrin* double mutants. Moreover, finger-like protrusions formed by the fusion-competent myoblast can only penetrate spectrin-free microdomains [52].

*Dynamin*: Dynamin catalyzes membrane fission by forming rings/helices around the neck of a budding endocytic vesicle [53]. In Haralalka et al. (2014) endocytotic vesicles were observed to be recruited to the site of fusion. These vesicles are responsible for the differential trafficking of Sns and Duf. However, the expression of dominant-negative Dynamin in myoblasts with *twist*-GAL4 alters myoblast specification and leads to an increased formation of founder cells [32]. The specification of founder cells involves Notch-mediated lateral inhibition, which requires Dynamin function [54].

A second role of Dynamin has been described during myoblast fusion in a more recent study. This study establishes Dynamin as a multifilament bundling protein [25] The capacity of forming actin bundles is based on the formation of a helical structure. Each Dynamin helix can engage 12–16 actin filaments. Actin binding involves the proline-rich domain of Dynamin. The disassembly of Dynamin helices involves GTP hydrolysis. The released Dynamin dimers and tetramers permit Arp2/3-based actin polymerization. The dynamin–actin interaction enhances mechanical strength of the actin network and is thought to propel invasive membrane protrusions during myoblast fusion.

In mammalian myoblast fusion, the scaffold protein Tks5 (tyrosine kinase substrate with five SH3 domains) has been demonstrated to interact with Dynamin-2 [55]. The RNAi mediated downregulation of Tks5 and the expression of dominant-negative Dynamin-2 decreases the fusion efficiency of myoblasts. Furthermore, electron microscopy studies showed that Dynamin-2 assembles around F-actin [55].

*Further proteins involved in signaling myoblast fusion:* In addition to described signaling molecules and F-actin polymerization regulators, further proteins have been identified during *Drosophila* myoblast fusion. However, their function is not as well understood. An adaptor protein that is exclusively expressed in founder cells is the multidomain protein Rolling pebbles 7/Antisocial (Rols7/Ants) [56,57,58]. The loss of *rols/ants* does not result in a complete loss of myoblast fusion. Instead, myoblasts are still able to form small syncytial cells. A maternal contribution of the *rols7/ants* mRNA is not responsible for syncytial formation. Rols7/Ants colocalizes with Duf in a ring-like manner and physically interacts with Duf [14,59]. The specific localization of Rols7/Ants is lost in mutants with impaired cell adhesion [57,58]. *Drosophila* Rols and its mammalian homologue TANC1 are targets of the Pax–Foxo1 transcription factor. The Pax–Foxo1 fusion gene is generated by a chromosomal translocation of Pax3 to Foxo1 (2;13) or by Pax7 to Foxo1 (1;13) and is the cause of skeletal muscle-lineage rhabdomyosarcoma [60].

Another protein that is expressed in during myoblast fusion in fusion-competent myoblasts and is located at the site of cell–cell contact is the calcium-binding protein Swiprosin-1 (Swip-1) [61]. The recruitment of Swip-1 to the plasma membrane depends on its EF-hand and a coiled–coiled domain. Swip-1 has been suggested to regulate the exocytosis of electron-dense vesicles of the pre-fusion complex.

The gene *singles bar* (*sing*) was isolated in a large *Drosophila* transcriptome analysis for primary mesodermal cells [19,62]. Singles bar is characterized by the presence of four transmembrane helixes and belongs to the protein family of MARVEL domain proteins. In vertebrates, members of this family are involved in tight junction formation and vesicle trafficking [63]. The *sing* mRNA is expressed in founder cells and fusion-competent myoblasts and myoblasts are still able to adhere in the absence of Sing. Electron microscopy studies on embryos suggest a requirement for Sing in the progression of the pre-fusion complex [19].

## 3. Common and Different Players in Visceral and Flight Muscle Development

### 3.1. The Formation of the Visceral Muscles during Embryogenesis

The *Drosophila* midgut is composed of visceral muscles that consist of an inner and outer layer of circular and longitudinal muscles. Circular and longitudinal muscles are interwoven and form a contractile network responsible for the contractility of the gut. Analogous to somatic musculature, muscles of the visceral mesoderm are composed of founder cells and fusion-competent myoblasts. However, the founder cells of the circular and longitudinal muscles arise from different areas of the mesoderm (Figure 5A). Circular visceral founder cells (cFCs) and fusion-competent myoblasts (cFCMs) originate from the visceral trunk mesoderm shortened as TVM [64,65]. The fate of the circular founder cells is determined by the spatial restriction of Jelly belly that binds to the Alk receptor tyrosine kinase along the visceral margin of the TVM [66,67,68]. The dorsally located cells become fusion-competent myoblasts (Figure 5B). A single circular founder cells fuses with one fusion-competent myoblasts. However, membrane fusion is incomplete and multiple cytoplasmatic bridges between the cells are formed.

In contrast, longitudinal founder cells arise from the caudal visceral mesoderm (CVM) and are characterized by the expression of the bHLH transcription factor HLH54F [69]. During midgut development, the longitudinal founder cells migrate along the visceral trunk mesoderm (Figure 5B,C, blue cells) and fuse with the remaining visceral fusion-competent myoblasts of the trunk mesoderm [64,68]. Studying the loss-of-function phenotype of genes required for somatic myoblast fusion has revealed that different genetic programs drive the fusion of circular visceral myoblasts and longitudinal visceral myoblasts [70]. These data are summarized in Figure 6. In comparison to somatic myoblast fusion, the fusion process leading to longitudinal muscle formation does not involve the function of Wip (Figure 5H) and WASp [70], but WASP-dependent Arp2/3-activation is clearly required for the migration of longitudinal founder cells (Figure 5I,J).

### 3.2. The Formation of Flight Muscles during Metamorphosis

The flight muscles share many morphological and developmental characteristics with skeletal muscles and, thus, serve as a compelling model for myoblast fusion [71]. The indirect flight muscles (IFM) of the thorax consist of dorsal longitudinal muscles (DLMs) and dorsoventral muscles (DVMs) [72]. The DLMs arise from three larval muscles (called larval dorsal oblique muscles = LOMs) and escape histolysis during metamorphosis [73,74,75]. These muscles serve as template muscles for the generation of DLMs. The myoblasts that fuse with these template muscles are first located as adult muscle progenitors (AMPs) at the wing imaginal disc (Figure 7). They are kept in an undifferentiated state by the expression of Notch. During larval development these cells start to proliferate and generate myoblasts [76]. This proliferation phase is characterized by the expression of Notch and Twist [77]. To subsequently initiate the muscle differentiation program, the expression of the Myocyte Enhancer factor 2 (Mef2) is increased [77,78]. Fusion occurs 12–20 h after pupal formation. During this process the three LOM template muscles split and give rise to six dorso-longitudinal muscles per hemithorax (Figure 7). The splitting of the template muscles is directly regulated by the fusion of myoblasts and a failure in fusion prevents LOM splitting.

Similar to larval myoblast fusion, the cell adhesion molecules Duf, Rst, and Sns are essential for myoblast-muscle template adhesion. Myoblasts that are close to the template muscles express Sns (Figure 7). This expression becomes restricted when the myoblast is contacted by filopodia extended by the template muscle (Figure 7). Once the myoblast is in close proximity to the template muscle cell membrane, Sns recruits Duf to the site of cell contact [79] (Figure 7). The fusion of myoblasts to template muscles, seem to require the same fusion-related genes as identified in somatic myoblast fusion [80,81]. Among these are the nucleation-promoting factors WASp and WAVE. In zygotic *wasp* mutants, myoblasts are capable to migrate from the wing imaginal disc to the DLM fibers, but since myoblast fusion is blocked completely, splitting does not occur in these mutants [80]. During embryonic somatic myoblast fusion actin-based mechanical forces have been proposed to drive membrane fusion [21]. Instead, in flight muscle development transmission electron microscopy studies show that actin forces are required for the flattening of the adhering myoblast and for the reduction of the membrane distances (<10 nm) between the fusing cells [81] (Figure 7).

Recently satellite-like cells have been identified in the flight musculature [82]. This cell population is marked by the zinc-finger homeodomain 1 (Zfh1) transcription factor. However, how do these cells elude the differentiation program during larval and adult development? Larva express the *zfh1*-long isoform in all adult AMPs. This isoform can be silenced by the microRNA mir-8 to facilitate AMP differentiation. The isoform is only expressed in a few AMPs and is insensitive to mir-8. The expression of *zfh1*-short is driven by a Notch responsive element [83]. The maintenance of Zfh1 in AMPs let them escape from differentiation.

### 3.3. Summary between Visceral and Flight Muscle Development

Taken together, studies on somatic and visceral myoblast fusion in the embryo and in flight muscle development during metamorphosis, have demonstrated the importance of the Duf–Sns-based adhesive interaction for myoblast fusion. A major difference between somatic, visceral and flight muscle development is the activation of the Arp2/3 complex. The nucleation-promoting factors WAVE and WASp are both required for Arp2/3-based actin polymerization in somatic and flight muscle development. However, WASp-dependent Arp2/3 activation only seems to be important for the migration of the longitudinal founder cells along the trunk mesoderm and not for the fusion of longitudinal founder cells with the fusion-competent myoblasts of the trunk mesoderm. The differences between embryonic longitudinal muscle and flight muscle formation are summarized in Table 1.

## 4. Myoblast Fusion in Mammals

Many of the described players in *Drosophila* myoblast fusion are conserved in mammals. Like in *Drosophila*, various cell adhesion molecules regulate the recognition and adhesion of myoblasts. Upon the interaction of myoblasts the fusion signal is translated into the cell, as summarized in Table 2 [34,84,85,86]. Recently, two muscle-specific transmembrane proteins have been identified in mammals called Myomerger (also known as Myomixer or Minion) and Myomaker [87,88,89,90]. Mutations in Myomaker cause rare congenital myopathy [86]. Myomaker and Myomerger act sequentially and control different steps of myoblast fusion. Whereas the lipid bilayers of the myoblasts lose their ability to hemifuse in homozygous *myomaker* mutants, homozygous *myomerger* mutants retain this ability. Instead, they fail to create a fusion pore. Interestingly, the block of actin polymerization also blocks Myomaker/Myomerger-based fusion [90]. The induction of hemifusion depends on the bilateral expression of Myomaker in myoblasts. In contrast, Myomerger function is required unilateral in C2C12 cells [90]. These findings suggest for the first time that the genetic composition of myoblasts in mammals differ like in *Drosophila*. It is a matter of debate whether de novo myoblast fusion requires the bilateral function of Myomaker and Myomerger and regeneration the unilateral function of Myomerger [85]. So far, no known homologues or structural similarities to fusogens have been identified. The transcription of Myomerger and Myomaker in humans is controlled by the muscle-specific transcription factor MyoD that binds to the E-box motif on the promotors of these genes [91]. In chicken, the activation of Myomaker further involves MyoG. In contrast, the miR-140-3p was identified to partially inhibit Myomaker expression [92].

## 5. Conclusions

Myoblast fusion is fundamental for muscle growth and repair after injury. Its alteration contributes to muscle diseases. The environmental cues during muscle regeneration by muscle stem cells must be tightly regulated to ensure the expansion of the muscle stem cell population, so that the damaged myofiber can be healed. Many epigenetic factors have been identified in muscle regeneration involving the activation, proliferation, and differentiation of muscle stem cells after injury. In particular, however, the mechanism governing myoblast fusion is incompletely understood, and very few epigenetic factors have been described. Ash1L (Kmt2h), a member of the Trithorax group, is the only epigenetic factor in mammalian myoblast fusion and regulates the cell adhesion protein Cdon by histone methylation [93]. Furthermore, Ash1L was found to be downregulated in Duchenne muscular dystrophy. With Myomerger and Myomaker, two proteins with fusogenic capacity have been identified.

In the past, *Drosophila* has been the dominant model organism to study myoblast fusion. Although the formation of the larval bodywall musculature serves as a convincing model to study basic principle of myoblast fusion, a major disadvantage of this experimental system was that these muscles do not regenerate. The identification of satellite-like stem cells in indirect flight muscles opens up new possibility to study regeneration and epigenetic reprogramming in *Drosophila*. Interestingly, the discovery of a possible unilateral requirement of Myomixer might hint that the identity of the myoblasts differs shortly before membrane fusion, which makes them more related to the concept of founder cells and fusion-competent myoblasts in *Drosophila*. How regeneration factors and fusogenes differ between *Drosophila* and mammals will be key questions for future studies. Indeed, the current stage of knowledge is that the basal fusion machinery from cell adhesion to actin polymerization is more or less conserved (Figure 8A,B). But the molecules and mechanisms driving the final process of membrane fusion differ (Figure 8A, red box).

## Figures and Tables

**Figure 1 cells-11-00321-f001:**
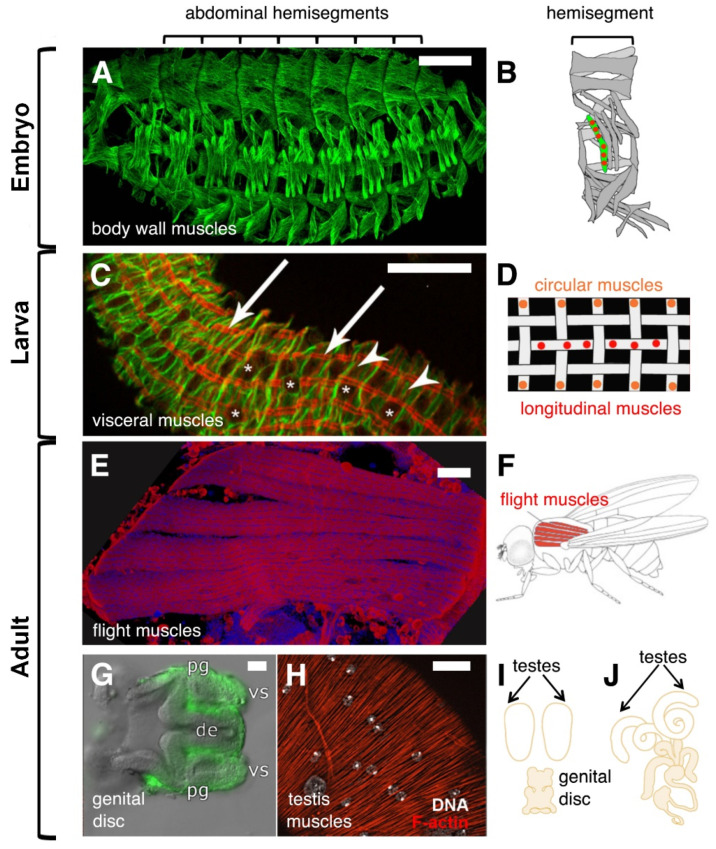
*Drosophila* syncytial muscles in embryos and adults. (**A**) Muscle pattern of a stage 16 wild-type embryo visualized by an antibody against β3-Tubulin. Scale bar 50 µm. (**B**) Schematic representation of the 30 multinucleated muscles within an abdominal hemisegment. (**C**) Midgut isolated from first instar larvae stained with phalloidin (**red**) and an extra cellular matrix marker (**green**). Arrows point to longitudinal muscles, asterisks to circular muscles. Asterisks indicate the position of the nuclei. Scale bar 20 µm. (**D**) Schematic representation of bi-nucleated circular (**orange**) and multinucleated longitudinal muscles (**red**). (**E**) The flight muscles called Dorsal Longitudinal Muscles (DLMs) that were stained with phalloidin (**red**) and DAPi (**blue**) are shown. Scale bar 50 µm. (**F**) Schematic image of the six DLMs. (**G**) Male genital disc 16 h after pupal formation (APF). The myoblasts that are located on the seminal vesicles (vs) of the genital disc are marked by the expression of UAS-mCD8-GFP. Scale bar 20 µm. Abbreviations: pg = paragonium and de = ejaculatory duct. (**H**) Arrangement of three-nucleated testis muscles stained with phalloidin (**red**). The nuclei of the muscles were visualized by using Hoechst dye and are shown in grey (**arrows**). Scale bar 20 µm. Large nuclei represent the nuclei of the overlaying pigment cells. (**I**) Schematic illustration of the genital disc and the testes. The genital disc contains a pool of myoblasts, which start to fuse on the genital disc 28 h APF. At this point of development, the paired testes are myoblast-free. (**J**) At 36 h APF, the testis and the seminal vesicles are fused. At 32 h APF myotubes start to migrate onto the testis until muscles surround it.

**Figure 2 cells-11-00321-f002:**
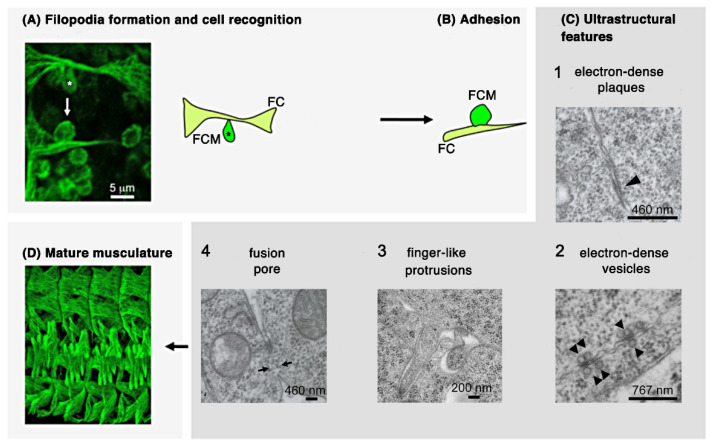
Scheme representing the cellular sequences of *Drosophila* myoblast fusion in the embryo. (**A**) Fusion-competent myoblasts (FCM; dark green) orient their filopodia towards the founder cell to which they attach (FC/growing myotube; light green). (**B**) Founder cells and fusion-competent cells align and come into close contact. (**C**) Electron microscopy studies revealed additional steps of myoblast fusion: (1) electron-dense plaques and, (2) vesicles, (3) formation of finger-like protrusions, and (4) fusion pores. The numbering (1) to (3) does not necessarily reflect the temporal appearance of these structures. (**D**) This leads finally to the formation of the larval multinucleated bodywall musculature.

**Figure 3 cells-11-00321-f003:**
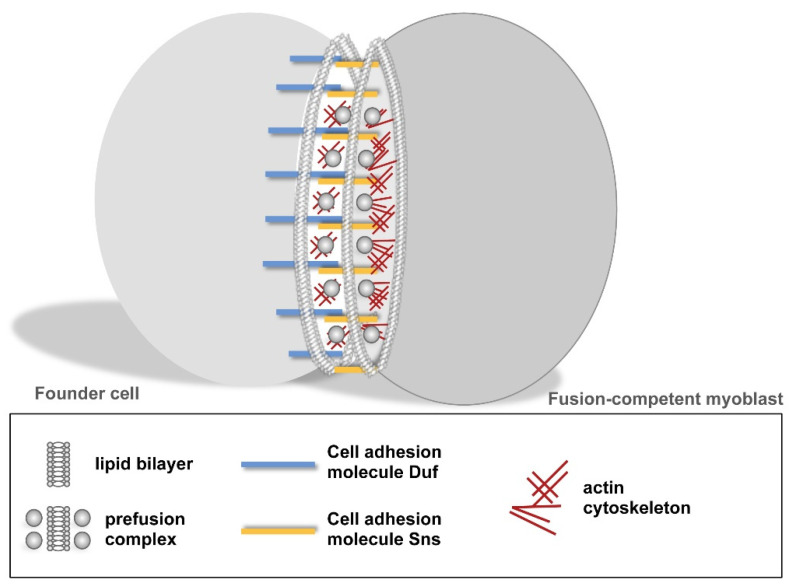
Formation of a cell–cell communication structure. The recognition and adhesion of myoblasts is established by Duf–Sns interactions. This process triggers a protein-machinery that leads to the recruitment of electron-dense vesicles and F-actin polymerization at the apposing membranes.

**Figure 4 cells-11-00321-f004:**
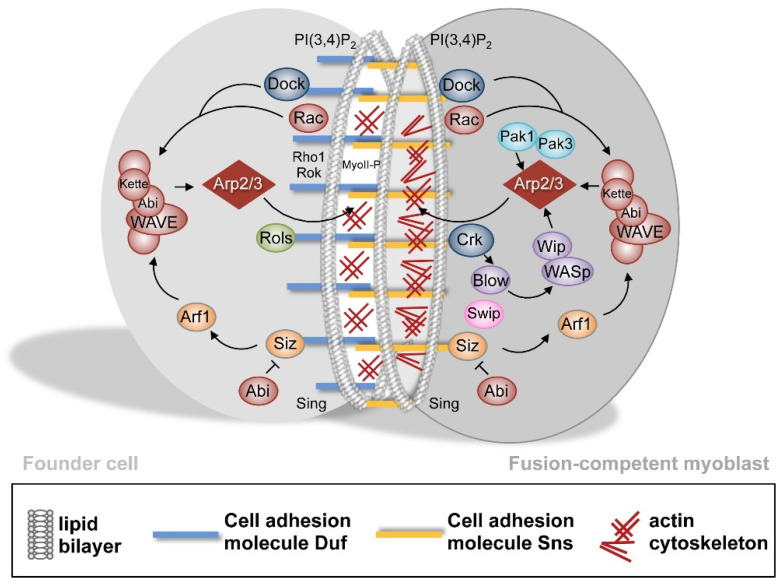
Intracellular signaling and actin polymerization during myoblast fusion. After the establishment of the Duf–Sns adhesive interaction, intracellular proteins are recruited myoblast-type specifically to the myoblast membrane. In response to WAVE activation in founder cells, the Arp2/3 nucleates a thin sheath of F-actin at the site of cell–cell contact. Additionally, the Rho1–Rok–MyoII pathway becomes activated in founder cells. In fusion-competent myoblasts, the activation of WAVE and WASp promotes Arp2/3-depemdent F-actin polymerization, which results in the formation of an F-actin-enriched focus.

**Figure 5 cells-11-00321-f005:**
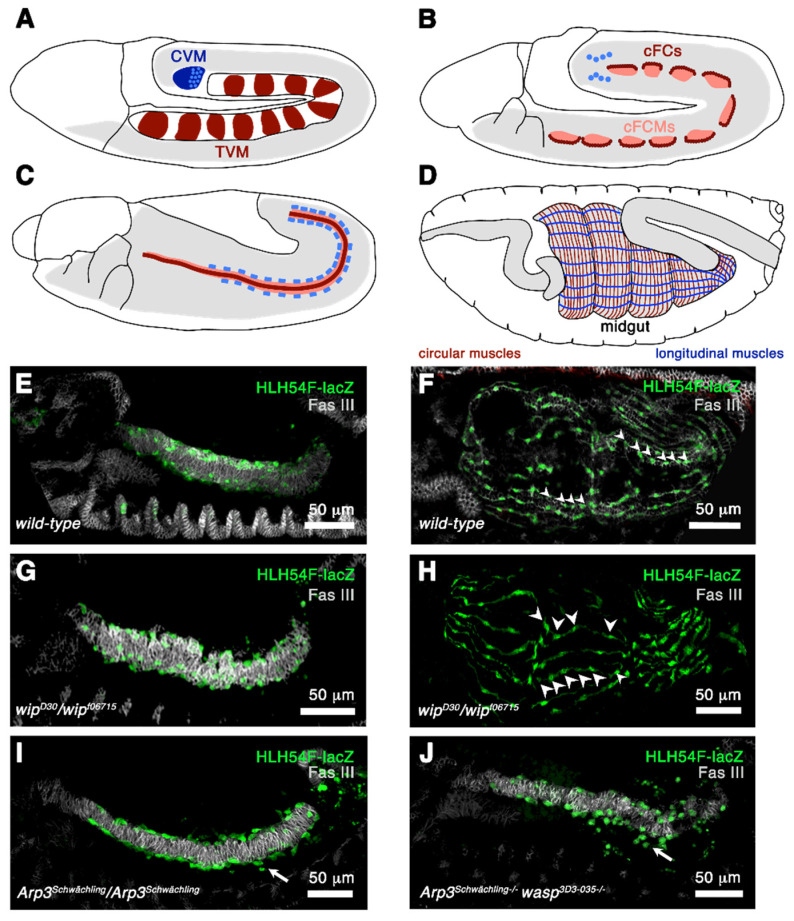
Formation of circular and longitudinal muscles. (**A**–**D**) Schematically drawn stages 10, 11, 12, and 16 embryos. Lateral view. (**A**) At stage 10 there are 11 cell clusters in the dorsal mesoderm that contain the primordial cells of the circular muscles and the longitudinal visceral muscles (**red**). These cells give rise to the trunk mesoderm (TVM). The caudal visceral mesoderm (CVM) gives rise to the longitudinal midgut muscles and is drawn in blue. (**B**) At stage 11, a single row of the visceral trunk primordia is specified and develops into founder cells (**red**). The rest of the cells evolve into fusion-competent myoblasts (**rose**). Additionally, the cells of the caudal visceral mesoderm leave their position at stage 11 and migrate onto the posterior cells of the trunk mesoderm on either side of the embryo. (**C**) At stage 12 the trunk visceral muscle founder cells fuse with fusion-competent myoblasts and form binucleated cells. The cells of the caudal visceral mesoderm are now circular founder cells and fuse with the fusion-competent myoblasts of the trunk mesoderm during migration. (**D**) Circular and longitudinal muscles are required for normal midgut constrictions. (**E**,**F**) Stages 13 and 16 embryos stained with anti-FAS III to visualize circular muscles (**grey**) and with anti-β-Galactosidase to mark caudal visceral founder cells expressing HLH54-lacZ. (**E**) Stage 13 embryo. Longitudinal founder cells have migrated on the trunk mesoderm. (**F**) Stage 16 embryo with multinucleated longitudinal muscles. Arrowheads point to the nuclei. (**G**) In *wip* homozygous mutants the migration of longitudinal founder cells and (**H**) myoblast fusion is not affected (**arrowheads**). (**I**) In homozygous *Arp3* mutants the longitudinal founder cells sometimes fail to migrate along the trunk mesoderm (**arrow**). (**J**) This phenotype is enhanced when homozygous *Arp3 wasp* double mutants.

**Figure 6 cells-11-00321-f006:**
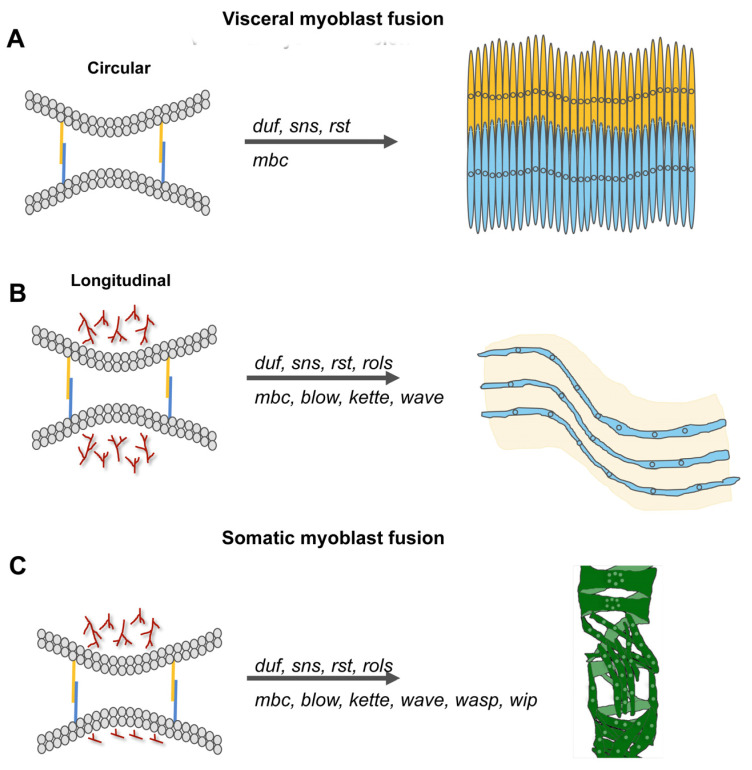
Molecular players of myoblast fusion increase with the complexity of the muscle that is formed. (**A**,**B**) Visceral muscles consist of binucleated circular muscle fibers that are interwoven with multinucleated longitudinal muscles. The fusion of founder cells and fusion-competent myoblasts during circular muscle formation is incomplete and depends on the cell adhesion molecules Duf, Rst, Sns, and the GEF Mbc. Longitudinal muscles contain up to six nuclei. The fusion of myoblasts is promoted by the presence of *duf, rst, sns, mbc, blow, rols,* and *kette*. (**C**) During somatic myoblast fusion, the fusion process is disturbed by the lack of *duf, sns, rst, rols, mbc, blow, kette, wave, wasp,* and *wip*.

**Figure 7 cells-11-00321-f007:**
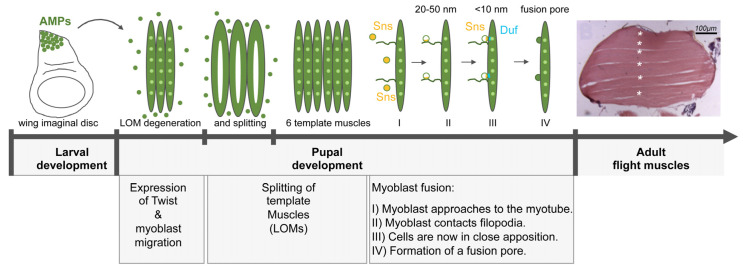
Model for IFM myoblast fusion. The larval oblique muscles (LOMs) that give rise to the DLM flight muscles partially degenerate 0–8 h after pupal formation (APF). The AMPs begin to migrate towards the LOMs and fuse to them. This induces the splitting of the LOMs at 12 h APF leading to the formation of six DLM fibers. (I) During IFM myoblast fusion, the myoblasts approach to the myotube and fusion-relevant genes like *sns* are switches on (yellow cells). (II) Next, the myoblasts make contact to the filopodia and Sns (yellow) is restricted to the contact site. The distance between the membranes is 20–50 nm. (III) The myoblast comes in close proximity to the myotube/template muscle and the cell adhesion molecule Duf (blue) is recruited to the site of cell–cell contact. The cells are now in close apposition (<10 nm). (IV) Finally, multiple fusion pores form, enlarge and the myoblast is incorporated into the myotube by the expansion of the fusion pore. Fusion is completed by 36 h APF.

**Figure 8 cells-11-00321-f008:**
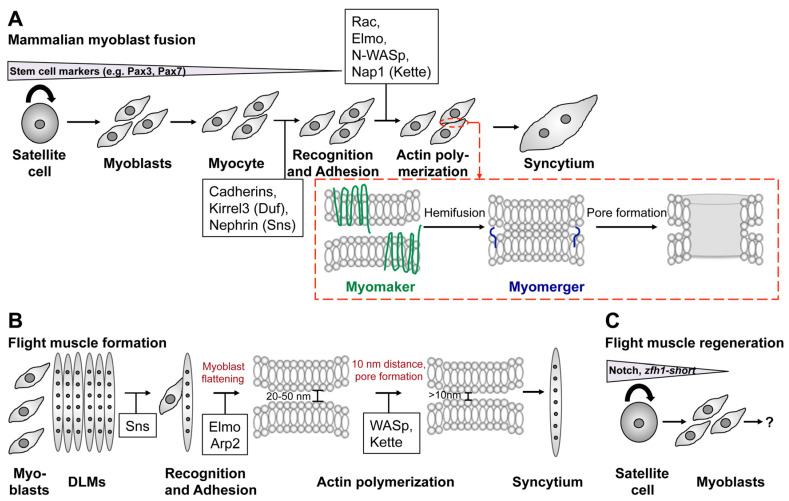
Common and different mechanisms in mammalian and *Drosophila* myoblast fusion. (**A**–**C**) Schematic representation of various steps in mammalian myoblast fusion (**A**) and *Drosophila* flight muscle development (**B**,**C**). (**A**) Many studies have used primary myoblasts or the murine C2C12 cell line to elucidate the role of Myomaker and Myomixer during myoblast fusion. Primary myoblasts derive by from satellite cells of skeletal muscles and are characterized by the expression of Pax7. Differentiation into myocytes and myotubes is achieved by the incubation in differentiation medium. Myoblast fusion involves recognition, adhesion, and actin cytoskeleton remodeling. Conserved players in this process are highlighted in the black boxes. Based on current knowledge Myomaker is responsible for inducing hemifusion (**red box**). Fusion pore formation involves the activity of Myomerger (**red box**). (**B**) During flight muscle development myoblasts migrate from the wing disc toward the DLM muscles and start to express Sns in the vicinity of the DLMs. The distance between the apposing membranes measures 20–50 nm. The Sns-expressing cell flattens, which requires the activity of Elmo and the Arp2/3 complex. WAVE- and WASp-dependent Arp2/3 activation brings the myoblast membranes in close proximity (<10 nm) and fusion pores are formed. (**C**) The identification of Zfh1-positive satellite stem cells opens new possibilities to study muscle regeneration in *Drosophila*.

**Table 1 cells-11-00321-t001:** List of proteins that are important for somatic myoblast fusion and play a role in longitudinal and/or DLM flight muscle formation.

	Embryo	Adult
	*Longitudinal muscles*	*Dorsal longitudinal muscles*
**Cell adhesion/Protein class**		
Kirre/Duf	required	required
Hbs	n.d.	required
Rst/IrreC	n.d.	required
Sing	n.d.	required
Sns	required	required
**Signaling molecules/Protein class**		
Blow (antagonist of WASp)	FCM	n.d.
Elmo (GEF)	n.d.	required
Mbc (GEF)	required	n.d.
Rols	required	n.d.
**Actin regulators/Protein class**		
Arp3	required	n.d.
Arp2	n.d.	required
Kette	required	required
Scar/WAVE	required	n.d.
WASp	Not required	required
Wip/Sltr	Not required	required

**Table 2 cells-11-00321-t002:** Proteins that function in cell adhesion, signaling, and actin regulation in mammals.

**Cell adhesion Proteins**	ADAM12, CD36, M-cadherin, N-cadherin, Disintegrin, α3-Integrin, α9-Integrin, β-Integrin, Kirrel3 (Duf homologue), Neogenin, Nephrin (Sns homologue)
**Transmembrane Lipids**	Cholesterol, Phosphatidyl serine
**Signaling Proteins**	Arf6, Brag-2, β-Catenin, Creatine kinase B, Crk, Crkl, diacylglycerol kinase ξ, Dock1, Dock5, EB3, focal adhesion kinase, Kindlin-2, Myoferlin, Stab2, Syntropin Trio
**Actin Proteins**	BAI1, BAI3, Cdc42, Filamin C, Nap1/Hem, Non-muscle myosin 2A, Rac1, N-WASP, Wip

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
