# Peer review of "Drosophila melanogaster: A Model System to Study Distinct Genetic Programs in Myoblast Fusion"

_cells, 2022, doi:10.3390/cells11030321_

Round 1

Reviewer 1 Report

This is an elegant review from an expert lab assembling large knowledge on myoblast fusion mechanisms in different types of muscles in Drosophila which is a reference model organism in the field. Comparison between mechanisms that underlie formation of embryonic body wall, visceral and adult flight and testis muscles provides originality to the paper and will be of interest to a large audience.

I have few suggestions that could further improve the quality of this review.

  1. The manuscript is illustrated by several nice figures. Among them Figure 1 is not particularly informative. I would suggest to either modify Fig.1 to a 3D-like view in which the rings of Duf and Sns but also F-actin distribution in FC and FCM are more realistic or skip it considering that the text provides all the information necessary to understand the basic picture of fusing cells. If you chose to keep/modify Figure 1, the legend need to be expanded.
  2. Figure 2 –please add scale bars to all panels. C) it shows 1st instar larvae, not embryo; E) specify that flight muscles shown are DLMs; G) « pg » and « de » are not defined.

  3. Figure 6 - the snapshot of embryonic muscle pattern is not necessary for this Figure dedicated to the adult flight muscles. I would also suggest to modify the scheme showing splitting of LOMs. Splitting requires myoblast fusion (LOMs do not split in Wasp mutants) so that it would be good to show that fusion promotes LOMs splitting. In the current version fusion is depicted only after splitting.
  4. Chapter 2.1.4 « Intracellular signaling and branched F-actin formation during myoblast fusion » is written in a way it is a catalog of fusion involved genes and their loss of function phenotypes. It could be replaced by a table or rearranged to a more narrative style.

  5. The chapter "Common and different players in visceral and flight muscle development" presents visceral muscle development and flight muscle formation but there is no comparison between these two developmental processes. This needs to be amended.

Minor points:

suggested modifications on yellow background

Line 83 –« largely destroyed » could be rather « undergo histolysis »

Line 84/85 – « In contrast to mammalian muscles, Drosophila muscles are built by one muscle fiber per muscle ». modify to : « In contrast to mammalian muscles, Drosophila body wall muscles are built by one muscle fiber per muscle ».

Line 97 – a set of larval muscles persists

Line 98 - serves as a template for development of a subset of flight muscles called Dorsal Longitudinal Muscles (DLM)

Line 107 – Figure 1 should be Figure 2

Line 147 – expresses

Line 148 – « How the reprogramming of the fusion-competent myoblast occurs not known yet ». modify to « How the reprogramming of the fusion-competent myoblast occurs is not known yet.

Line 191 – Figure 3 instead of Figure 1

Author Response

Response to Reviewer 1:

  1. The manuscript is illustrated by several nice figures. Among them Figure 1 is not particularly informative. I would suggest to either modify Fig.1 to a 3D-like view in which the rings of Duf and Sns but also F-actin distribution in FC and FCM are more realistic or skip it considering that the text provides all the information necessary to understand the basic picture of fusing cells. If you chose to keep/modify Figure 1, the legend need to be expanded.

We have deleted Fig.1 from the „Introduction“ and have added a modified version oft he figure to chapter 2.1.3. The modified figure is now Fig. 3.

  1. Figure 2 –please add scale bars to all panels. C) it shows 1st instar larvae, not embryo; E) specify that flight muscles shown are DLMs; G) « pg » and « de » are not defined.

We have added the scale bars to the figure and revised the figure legnd as follow:

Figure 2. Drosophila syncytial muscles in embryos and adults. (a) Muscle pattern of a stage 16 wild-type embryo visualized by an antibody against b3-Tubulin. Scale bar 50 mm (b) Schematic representation of the 30 multinucleated muscles within an abdominal hemisegment. (c) Midgut isolated from 1st instar larvae stained with phalloidin (red) and an extra cellular matrix marker (green). Arrows point to longitudinal muscles, arrowheads to circular muscles. Asterisks indicate the position of the nuclei. Scale bar 20 mm. (d) Schematic representation of bi-nucleated circular (orange) and multinucleated longitudinal muscles (red). (e) The flight muscles called Dorsal Longitudinal Muscles (DLMs) that were stained with phalloidin (red) and DAPi (blue) are shown. Scale bar 50 mm. (f) Schematic image of the six DLMs. (g) Male genital disc 16 h after pupal formation (APF). The myoblasts that are located on the seminal vesicles (vs) of the genital disc are marked by the expression of UAS-mCD8-GFP. Scale bar 20 mm. Abbreviations: pg=paragonium and de=ejaculatory duct (h) Arrangement of three-nucleated testis muscles stained with phalloidin (red). The nuclei of the muscles were visualized by using Hoechst dye and are shown in grey (arrows). Scale bar 20 mm. Large nuclei represent the nuclei of the overlaying pigment cells. (i) Schematic illustration of the genital disc and the testes. The genital disc contains a pool of myoblasts, which start to fuse on the genital disc 28 h APF. At this point of development, the paired testes are myoblast-free. (j) At 36 h APF, the testis and the seminal vesicles are fused. At 32 h APF myotubes start to migrate onto the testis until muscles surround it.”

Furthermore we have added “larval midgut muscles” to chapter 2:

The larval midgut muscles are composed of binucleated circular muscles that show an incomplete fusion and of multinucleated longitudinal muscles.”

  1. Figure 6 - the snapshot of embryonic muscle pattern is not necessary for this Figure dedicated to the adult flight muscles. I would also suggest to modify the scheme showing splitting of LOMs. Splitting requires myoblast fusion (LOMs do not split in Wasp mutants) so that it would be good to show that fusion promotes LOMs splitting. In the current version fusion is depicted only after splitting.

We have modified Fig. 6 and have deleted the embryo in the figure. The three template muscles with the white area should depict the splitting LOMs. These muscles are now surrounded by myoblasts and we labeled the process shown in the figure as LOMs degradation and splitting. Fig. 6 is Fig. 7 in the revised manuscript. Additionally we modified the text in chapter 3.2:

During this process the three LOM template muscles split and give rise to six doro-longitudinal muscles per hemithorax (Figure 7). The splitting of the template muscles is directly regulated by the fusion of myoblasts and a failure in fusion prevents LOM splitting. (….)Zygotic wasp mutants are capable to migrate from the wing imaginal disc to the DLM fibers, but since myoblast fusion is blocked completely, splitting does not occur in these mutants [80]. During embryonic somatic myoblast fusion actin-based mechanical forces have been proposed to drive membrane fusion [21]. Instead, in flight muscle development transmission electron microscopy studies show that actin forces are required for the flattening of the adhering myoblast and for the reduction of the membrane distances (< 10 nm) between the fusing cells [81] (Figure 8).

  1. Chapter 2.1.4 « Intracellular signaling and branched F-actin formation during myoblast fusion » is written in a way it is a catalog of fusion involved genes and their loss of function phenotypes. It could be replaced by a table or rearranged to a more narrative style.

Instead of a table, we have chosen to add a figure (Fig. 4) to this chapter.

  1. The chapter "Common and different players in visceral and flight muscle development" presents visceral muscle development and flight muscle formation but there is no comparison between these two developmental processes. This needs to be amended.

Indeed a comparison between visceral and flight muscle formation was missing and we have added now a summary as chapter 3.3 which includes a table:

3.3 Summary between visceral and flight muscle development

Taken together, studies on somatic and visceral myoblast fusion in the embryo and in flight muscle development during metamorphosis, have demonstrated the importance of the Duf-Sns-based adhesive interaction for myoblasts fusion. A major difference between somatic, visceral and flight muscle development is the activation of the Arp2/3 complex. The nucleation-promoting factors WAVE and WASp are both required for Arp2/3-based actin polymerization in somatic and flight muscle development. However, WASp-dependent Arp2/3 activation only seems to be important for the migration of the longitudinal founder cells along the trunk mesoderm and not for the fusion of longitudinal founder cells with the fusion-competent myoblasts of the trunk mesoderm. The differences between embryonic longitudinal muscle and flight muscle formation are summarized in Table 1.

Table 1 - List of proteins that are important for somatic myoblast fusion and play a role in longitudinal and/or DLM flight muscle formation“

Minor points were changed as suggested by the reviewer.

Reviewer 2 Report

In paragraph 1. Introduction, authors discussed the formation of cell-cell communication structure. The structure was shown in Fig1. However, the figure legend is missing and there is no explanation of the abbreviations of involved molecules in the figure.

In paragraph 2.1.3 Establishment of a cell-cell communication structure after the recognition and adhesion of myoblasts. The authors described the molecules involved in the formation of this communication structure.  Fig. 1 could be moved here and details of this communication structure should be described.  In 1 introduction, it will be better only to discuss the overall process without going to details.

In Fig. 3 labeling in the Figure does not match what described in the Figure legend.

In paragraph 4. Myoblast fusion in mammals 

It will be helpful to have a figure to summarize the similarity and difference regarding the steps of myoblast fusion between mammalian skeletal muscle and Drosophila flight muscle.

Author Response

Response to Reviewer 2:

1a) In paragraph 1. Introduction, authors discussed the formation of cell-cell communication structure. The structure was shown in Fig1. However, the figure legend is missing and there is no explanation of the abbreviations of involved molecules in the figure.

1b) In paragraph 2.1.3 Establishment of a cell-cell communication structure after the recognition and adhesion of myoblasts. The authors described the molecules involved in the formation of this communication structure. Fig. 1 could be moved here and details of this communication structure should be described. In 1 introduction, it will be better only to discuss the overall process without going to details.

We have deleted this figure from the introduction and have added it to chapter 2.1.3 as suggested. We have added the following figure legend to this new Fig. 3:

Figure 3 Formation of a cell–cell communication structure. The recognition and adhesion of myoblasts is established by Duf-Sns interactions. This process triggers a protein-machinery that leads to the recruitment of electron-dense vesicles and F-actin polymerization at the apposing membranes.”

2) In Fig. 3 labeling in the Figure does not match what described in the Figure legend.

We apologize for this mistake and have changed now the order of the figure legend, which now matches the shown images. Fig. 3 is Fig.2 in the revised version of the manuscript.

Figure 2. Scheme representing the cellular sequences of Drosophila myoblast fusion in the embryo. (a) Fusion-competent myoblasts (FCM; dark) orient their filopodia towards the founder cell to which they attach (FC/growing myotube; light green). (b) Founder cells and fusion-competent cells align and come into close contact. (c) Electron microscopy studies revealed additional steps of myoblast fusion: (1) electron-dense plaques and, (2) vesicles, (3) formation of finger-like protrusions and (4) fusion pores. The numbering (1) to (3) does not necessarily reflect the temporal appearance of these structures. (d) This leads finally to the formation of the larval multinucleated bodywall musculature.”

3) In paragraph 4. Myoblast fusion in mammals

It will be helpful to have a figure to summarize the similarity and difference regarding the steps of myoblast fusion between mammalian skeletal muscle and Drosophila flight muscle.

We designed a new figure that highlights common and different molecules in both processes. This new figure has been labeled as Fig. 8.